# Prevalence and Significance of Hypermetabolic Lymph Nodes Detected by 2-[^18^F]FDG PET/CT after COVID-19 Vaccination: A Systematic Review and a Meta-Analysis

**DOI:** 10.3390/ph14080762

**Published:** 2021-08-03

**Authors:** Giorgio Treglia, Marco Cuzzocrea, Luca Giovanella, Luigia Elzi, Barbara Muoio

**Affiliations:** 1Clinic of Nuclear Medicine, Imaging Institute of Southern Switzerland, Ente Ospedaliero Cantonale, 6500 Bellinzona, Switzerland; marco.cuzzocrea@eoc.ch (M.C.); luca.giovanella@eoc.ch (L.G.); 2Department of Nuclear Medicine and Molecular Imaging, Lausanne University Hospital, 1011 Lausanne, Switzerland; 3Academic Education, Research and Innovation Area, General Directorate, Ente Ospedaliero Cantonale, 6500 Bellinzona, Switzerland; 4Faculty of Biology and Medicine, University of Lausanne, 1011 Lausanne, Switzerland; 5Faculty of Biomedical Sciences, Università della Svizzera Italiana, 6900 Lugano, Switzerland; 6Clinic for Nuclear Medicine, University Hospital and University of Zürich, 8091 Zürich, Switzerland; 7Department of Medicine, Ente Ospedaliero Cantonale, 6500 Bellinzona, Switzerland; luigia.elzi@eoc.ch (L.E.); barbara.muoio@eoc.ch (B.M.); 8Faculty of Medicine, University of Basel, 4056 Basel, Switzerland

**Keywords:** COVID-19, vaccine, PET, FDG, lymph nodes, inflammation, systematic review, meta-analysis

## Abstract

Recently, several articles reported incidental findings at 2-[^18^F]FDG PET/CT in patients who have received COVID-19 vaccinations, including hypermetabolic axillary lymph nodes (HALNs) ipsilateral to the COVID-19 vaccine injection site which may cause diagnostic dilemmas. The aim of our work was to calculate the prevalence of this finding. A comprehensive computer literature search of PubMed/MEDLINE, Embase, and Cochrane library databases was performed to identify recently published articles that investigated the prevalence of HALNs detected by 2-[^18^F]FDG PET/CT after COVID-19 vaccination. Pooled prevalence of this finding was calculated through a meta-analytic approach. Nine recently published articles including 2354 patients undergoing 2-[^18^F]FDG PET/CT after recent COVID-19 vaccination have been included in the systematic review. Overall, HALNs ipsilateral to the vaccine injection site were frequent findings mainly due to vaccine-related immune response in most of the cases. The pooled prevalence of HALNs after COVID-19 vaccination was 37% (95% confidence interval: 27–47%) but with significant heterogeneity among the included studies. Physicians must be aware and recognize the significant frequency of HALNs at 2-[^18^F]FDG PET/CT related to immune response to vaccine injection. Larger studies are needed to confirm the findings of this systematic review and meta-analysis.

## 1. Introduction

Since the beginning of 2020, several vaccines that can control the COVID-19 pandemic have been developed [1,2]. These vaccines, including messenger ribonucleic acid (mRNA), adenoviral-vectored, protein-subunit, and whole-cell inactivated-virus vaccines, have now reported efficacy in phase III trials, receiving approval in many countries [1,2]. Most COVID-19 vaccines are designed to elicit immune responses against the SARS-CoV-2 spike protein [1,2].

Due to their immunological effects, vaccinations including those related to COVID-19 may cause various confusing imaging patterns that pose diagnostic challenges for clinicians and pitfalls for reading radiologists and nuclear medicine physicians [3,4,5,6]. On the other hand, some imaging methods could be of potential value to assess the immune response after vaccination, including those related to COVID-19 [3,4].

Fluorine-18 fluorodeoxyglucose (2-[^18^F]FDG) positron emission tomography/computed tomography is a hybrid imaging technique used for several indications including the evaluation of oncological and infectious/inflammatory diseases [7]. 2-[^18^F]FDG is a radiolabeled glucose analogue; this radiopharmaceutical is taken up by cells via cell membrane glucose transporters and subsequently phosphorylated with hexokinase inside most cells. The ability of 2-[^18^F]FDG PET/CT to identify tumor lesions or sites of inflammation and infection is mainly related to the glycolytic activity of the neoplastic or inflammatory cells [7].

Incidental lesions detected by whole-body 2-[^18^F]FDG PET/CT performed for several indications are widely described in the literature [8]. Since the COVID-19 vaccination programs began worldwide, several articles (mainly case reports) reported incidental findings at 2-[^18^F]FDG PET/CT in patients who have received recent COVID-19 vaccinations [5,6,9]. These findings due to vaccine-related immune response are mainly represented by hypermetabolic axillary lymph nodes (HALNs) ipsilateral to the COVID-19 vaccine injection site (a case example is reported in Figure 1) and 2-[^18^F]FDG uptake in the vaccine injection site [5,6,9].

As these incidental findings at 2-[^18^F]FDG PET/CT may create several diagnostic dilemmas for radiologists and nuclear medicine physicians, in particular in discriminating between neoplastic and inflammatory lesions, the aim of this work was to perform a systematic review and meta-analysis of the existing literature to calculate their prevalence.

## 2. Results

### 2.1. Literature Rearch

A comprehensive computer literature search based on the review question and using PubMed/MEDLINE, Embase, and Cochrane library databases retrieved 70 records. Original articles reporting data about the prevalence of HALNs detected by 2-[^18^F]FDG PET/CT after COVID-19 vaccination and including more than 10 patients were considered eligible for inclusion in this systematic review. Nine original articles were selected and retrieved in full-text version and 61 records were excluded. The detailed process of article selection is reported in Figure 2. There were no additional articles found after a further screening of the references of those selected articles. Thus, nine studies including 2354 patients were included in the qualitative and quantitative analyses [10,11,12,13,14,15,16,17,18]. The characteristics of the studies included in this systematic review are shown in Table 1, Table 2 and Table 3.

### 2.2. Qualitative Synthesis (Systematic Review)

Nine recently published articles including 2354 patients undergoing 2-[^18^F]FDG PET/CT after recent COVID-19 vaccination were included in the systematic review [10,11,12,13,14,15,16,17,18]. Patient and basic studies characteristic are reported in Table 1.

All the articles were retrospective monocentric studies with mild quality according with the NIH National Heart, Lung, and Blood Institute Study Quality Assessment Tools. Five out of nine studies were carried out in Israel, two in USA, one in Korea and one in Switzerland.

With respect to patient characteristics, most of the patients included in the selected articles underwent 2-[^18^F]FDG PET/CT for oncological indications. Mean age largely varied among the studies, from 45 to 76 years. The male sex ratio also varied among the included studies between 35% and 72%.

With respect to the type of COVID-19 vaccination, mRNA vaccination was performed in eight out nine studies whereas adenovirus vectored vaccine was carried out in one study only. The time between COVID-19 vaccination and 2-[^18^F]FDG PET/CT largely varied among the studies from one to 71 days. Furthermore, in cases of COVID-19 vaccines administered in two doses, some patients underwent 2-[^18^F]FDG PET/CT after the first dose (from 34% to 80% of cases) and others after the second dose (from 20% to 100% of cases).

As shown in Table 2, technical aspects of 2-[^18^F]FDG PET/CT largely varied among the included studies.

Overall, all the studies reported that HALNs ipsilateral to the vaccine injection site were present in a significant percentage of patients undergoing 2-[^18^F]FDG PET/CT after COVID-19 vaccination (Table 3) and they were considered as a consequence of vaccine-related immune response in most of the cases. The percentage of HALNs detected by 2-[^18^F]FDG PET/CT after COVID-19 vaccine ranged from 7% to 90% of cases; 2-[^18^F]FDG uptake at the vaccine injection site was also frequently reported (from 12% to 70% of cases). In most of the studies, an increase of percentage of HALNs detected by 2-[^18^F]FDG PET/CT after the second dose of vaccine compared to the first dose was reported. Even if there was not difference in number, size, or 2-[^18^F]FDG uptake of HALNs among the two groups, after the booster vaccine the incidence of high-intensity HALN was higher than after the first vaccine, and so was the size of nodes, detection of hypermetabolic lymph nodes beyond the ipsilateral axillary region (e.g., supraclavicular and cervical lymph nodes), and detection of the vaccination site.

Most of HALNs detected by 2-[^18^F]FDG PET/CT were not enlarged at CT scan (<10 mm short axis). HALNs were more frequent in immunocompetent than in immunocompromised patients; furthermore, these findings were more frequent and with higher 2-[^18^F]FDG uptake in younger than in older patients (>64 years old).

With respect to the time points, the incidence and grade of HALNs were highest during the first weeks after COVID-19 vaccine and decreased gradually over time. A significant percentage of these findings was even observed more than six weeks after vaccination with persistence even at 10 weeks after COVID-19 vaccination.

In patients with hematological malignancy, HALNs detected on 2-[^18^F]FDG PET/CT positively correlated with antibody-mediated immune response to COVID-19 vaccine.

In those studies comparing different COVID-19 vaccinations, HALNs at 2-[^18^F]FDG PET/CT were more frequent after Moderna than after Pfizer vaccine.

Even if HALNs ipsilateral to the vaccine injection site were considered as vaccine-related reactive lymph nodes in most of the cases, the nature of the HALNs may have been equivocal in a minority of cases and malignancy cannot be excluded on PET/CT image analysis only (i.e., women with breast cancer ipsilateral to the vaccination arm, lymphoma patients with nodal disease above the diaphragm, and patients with upper limb sarcoma, melanoma, or head and neck malignancy with extensive nodal involvement). Additional focused sonography of the region and fine-needle aspiration were suggested as useful in these cases to exclude malignancy.

### 2.3. Quantitative Synthesis (Meta-Analysis)

Results of the quantitative analysis are reported in Table 3. The pooled prevalence of HALNs ipsilateral to vaccine injection site and detected by 2-[^18^F]FDG PET/CT after recent COVID-19 vaccination was 37% with 95% confidence interval values (95%CI) ranging from 27% to 47%. Pooled prevalence of 2-[^18^F]FDG uptake in the injection site after COVID-19 vaccination was 30% (95%CI: 20–41%). A significant heterogeneity across studies was found by the I^2^ index whereas a significant publication bias was excluded by the Egger’s test.

Subgroup analyses taking into account first and second dose of COVID-19 vaccine demonstrated a higher prevalence of HALN after the second dose compared to the first dose (41% vs. 26%) even if this difference was not statistically significant due to the partial overlap of 95%CI values among these groups. Unfortunately, due to the limited number of available articles, other subgroup analyses to statistically explore the heterogeneity could not be performed (i.e., considering different types of COVID-19 vaccine or different time points after COVID-19 vaccination).

## 3. Discussion

During the SARS-CoV-2 pandemic several 2-[^18^F]FDG PET/CT imaging findings associated with COVID-19 were described [19,20,21]. Further 2-[^18^F]FDG PET/CT findings were described after the introduction of COVID-19 vaccines [5,6,9], in particular HALNs ipsilateral to the vaccine injection site. Our systematic review and meta-analysis clearly demonstrates that this 2-[^18^F]FDG PET/CT finding is frequent after recent COVID-19 vaccination and it is due to immune response to the vaccine in most of the cases [10,11,12,13,14,15,16,17,18]. Recognition of the vaccine-related HALNs may avoid patient anxiety and unnecessary further examinations or biopsies. These results seem to be in line with those of previous studies reporting a significant prevalence of HALNs after recent vaccinations beyond those related to COVID-19 (Table 4) [22,23,24,25,26,27,28,29].

The high frequency of HALNs at 2-[^18^F]FDG PET/CT after recent COVID-19 vaccination could be explained by the high sensitivity of this imaging method for detection of inflammatory abnormalities and immune response including reactive lymph nodes; cells involved in inflammation and immune response are able to take up 2-[^18^F]FDG due to their expression of high levels of glucose transporters and high hexokinase activity [7].

HALNs at 2-[^18^F]FDG PET/CT after recent COVID-19 vaccination were not enlarged in the majority of cases. This is not surprising as functional/metabolic abnormalities detected by 2-[^18^F]FDG PET usually precede morphological abnormalities detected by CT scan and other conventional imaging methods [7]. Furthermore, the pooled prevalence of HALNs detected by 2-[^18^F]FDG PET/CT was higher than the prevalence of axillary lymphadenopathies (short axis >10 mm) detected by conventional imaging techniques after COVID-19 vaccinations [5,6,9].

There is a trend towards a higher prevalence of HALNs at 2-[^18^F]FDG PET/CT after the second vaccine dose compared to the first dose; this finding can be explained by the boost reactivation after the second vaccine injection [10,11,12,13].

Notably, hypermetabolic lymph nodes at 2-[^18^F]FDG PET/CT after COVID-19 vaccination may also be detected outside the axillary region [10,11,12,13,14,15,16,17,18].

HALNs are more frequent in immunocompetent than in immunocompromised patients and more frequent in young than in older patients (<64 years), therefore, immune competency and age may play a role in the prevalence of this finding [10,11,12,13,14,15,16,17,18].

Although the prevalence of HALNs decreases over time after COVID-19 vaccination, HALNs at 2-[^18^F]FDG PET/CT were even observed 10 weeks after COVID-19 vaccination [15]. COVID-19 vaccines seem to be more immunostimulatory as compared to other traditional vaccine agents [1,2], and this may account for their longer-lasting vaccine-related HALNs. However, more studies are needed to clarify the time required after COVID-19 vaccination to allow for resolution of HALNs.

Another frequent finding at 2-[^18^F]FDG PET/CT after COVID-19 vaccination is the presence of radiopharmaceutical uptake in the vaccine injection site (deltoid muscle). This finding in cases of concomitant ipsilateral HALNs at 2-[^18^F]FDG PET/CT might be useful for determining the vaccine-related nature of HALNs. Whereas HALNs might be a marker of vaccine-induced immune system activation, increasing over time and after the second vaccine dose, the increased metabolic activity in the vaccine injection site is inflammatory in nature, and likely secondary to local trauma [10,11,12,13,14,15,16,17,18].

Even if the majority of HALNs detected by 2-[^18^F]FDG PET/CT after COVID-19 vaccination are related to immune response after vaccine injection, in some patients with oncological diseases, to discriminate HALNs due to COVID-19 vaccination from malignancy, further examinations such as ultrasound and biopsies would be necessary, especially in patients with breast cancer, lymphoma, or melanoma [10,11,12,13,14,15,16,17,18]. Because there is an overlap in the intensity of 2-[^18^F]FDG uptake among benign and malignant HALNs, this parameter cannot be used to discriminate these two entities after COVID-19 vaccination; conversely, information about tumor type, disease history, and previous imaging are very useful for assessing the nature of HALNs detected by 2-[^18^F]FDG PET/CT after COVID-19 vaccination.

Based on the potential risk of HALNs after COVID-19 vaccine to mimic or mask malignant disease at 2-[^18^F]FDG PET/CT, patients with cancer with a propensity for spread to ipsilateral axillary lymph nodes (e.g., breast cancer, melanoma, and lymphomas) should have the COVID-19 vaccine in the contralateral arm to the tumor site. Nuclear medicine technologists or anyone collecting anamnestic data should document vaccine site, date, type, and first versus second dose. Overall, recent ipsilateral COVID-19 vaccination history should be recognized as a potential common cause of HALNs on PET/CT scans.

Our systematic review and meta-analysis has several limitations. First, only a limited number of original articles related to the review question were available and all were retrospective monocentric studies with possible selection bias. Histologic proof to exclude malignancy in HALNs detected by 2-[^18^F]FDG PET/CT was obtained only in few cases, therefore a verification bias cannot be excluded. The included studies were heterogeneous for characteristics of patients included, type of COVID-19 vaccine, time between COVID-19 vaccination and 2-[^18^F]FDG PET/CT, and technical aspects of 2-[^18^F]FDG PET/CT. Due to limited data, only subgroup analyses taking into account HALNs after first and second dose of COVID-19 vaccine were performed. Other subgroup analyses (i.e., based on the different COVID-19 vaccine or the different time points after vaccination) to statistically explore the heterogeneity could not be performed. On the other hand, a significant publication bias was not observed in our analysis.

Even if described in the literature, we did not consider in our analysis the increased radiopharmaceutical uptake in axillary lymph nodes after COVID-19 vaccination using other PET radiopharmaceuticals beyond 2-[^18^F]FDG [14,16], due to their different uptake mechanisms and significance compared to 2-[^18^F]FDG. In addition, 2-[^18^F]FDG PET/CT findings reported in few studies such as splenic or non-axillary lymph nodes 2-[^18^F]FDG accumulation after COVID 19 vaccination [30] were not investigated in the present meta-analysis due to lack of sufficient data.

We did not focus our systematic review and meta-analysis on CT findings; the reason is that HALNs are frequently normal in size at CT scan [10,11,12,13,14,15,16,17,18]. However, we would like to underline that it is important that radiologists consider recent COVID-19 vaccination in the differential diagnosis of unilateral axillary lymphadenopathy and are aware of typical appearances of this findings across all imaging methods [31,32].

Larger studies are warranted to confirm the findings reported in this systematic review and meta-analysis. In particular the causal association between HALNs and immunogenicity elicited by COVID-19 vaccination should be further explored. Furthermore, more information about the trend and change of HALNs over time should be obtained. These data could be collected in future studies.

## 4. Materials and Methods

This systematic review conforms to the statement on “Preferred Reporting Items for Systematic Reviews and Meta-Analyses of Diagnostic Test Accuracy studies” (PRISMA-DTA) and to specific guidelines on systematic reviews and meta-analyses of diagnostic studies [33,34,35].

### 4.1. Search Strategy

A comprehensive computer literature search of PubMed/MEDLINE, Embase, and Cochrane library databases was performed independently by two authors (GT and MC) in order to identify recently published articles that investigated the prevalence of HALNs detected by 2-[^18^F]FDG PET/CT after COVID-19 vaccination. A search string was created taking into account the review question. The full search string used is reported in Appendix A. Moreover, further screening of the references of the selected articles was performed to search for additional studies.

The literature search was updated until 28 June 2021.

### 4.2. Study Selection

Original articles reporting data about the prevalence of hypermetabolic lymph nodes detected by 2-[^18^F]FDG PET/CT after COVID-19 vaccination were considered eligible for inclusion in this systematic review. The exclusion criteria were as follow: (a) articles outside of the field of interest of this review; (b) case reports and small case series (fewer than 10 cases) related to the review question; and (c) review articles, editorials, comments, letters, and conference proceedings related to the review question. No language or date restrictions were used. Nevertheless, only articles published in the last year were expected to be selected due to the recent availability of COVID-19 vaccines.

The titles and abstracts of the retrieved records were reviewed independently by two researchers (GT and MC) according to the inclusion and exclusion criteria previously mentioned. Articles which appeared evidently ineligible were rejected. The full-length versions of the remaining articles were independently reviewed by three researchers (GT, MC, and BM) to evaluate their eligibility for inclusion. Any disagreements over articles eligibility were resolved during an online consensus meeting.

### 4.3. Data Extraction

Two authors performed the data extraction from the eligible studies (GT and BM).

Basic study and patients data were collected for each eligible study including authors’ names, publication date, country, study design, type and number of patients recruited, patients’ age and sex, type of COVID-19 vaccine injected, and time between vaccine administration and PET/CT scan.

Technical aspects of 2-[^18^F]FDG PET/CT scans in the included studies were also extracted including data on PET/CT scanner and acquisition, 2-[^18^F]FDG activity injected, time between 2-[^18^F]FDG injection and PET/CT image acquisition, and PET/CT image analysis.

Lastly, data on the prevalence of HALNs at 2-[^18^F]FDG PET/CT after COVID-19 vaccine administration and data on the 2-[^18^F]FDG uptake at the vaccine injection site were extracted.

### 4.4. Quality Assessment

The overall quality of the studies included in this systematic review was appraised based on the NIH National Heart, Lung and Blood Institute Study Quality Assessment Tools [36].

### 4.5. Statistical Analysis

For each included study we calculated the prevalence of HALNs at 2-[^18^F]FDG PET/CT after COVID-19 vaccine as the ratio between the number of patients with HALNs detected by 2-[^18^F]FDG PET/CT after COVID-19 vaccine and the number of patients undergoing 2-[^18^F]FDG PET/CT after COVID-19 vaccine. A similar calculation was performed about the prevalence of 2-[^18^F]FDG uptake at the vaccine injection site for the same patients. When applicable, subgroup analyses were performed about prevalence of HALNs at 2-[^18^F]FDG PET/CT after first and second dose of COVID-19 vaccine.

Pooled prevalence values were calculated through a meta-analytic approach which considered the weight of each included study. A random-effects model was used for statistical pooling of the data taking into account the heterogeneity between studies. Pooled data were presented with their respective 95% confidence interval (95% CI) values.

Inconsistency index (I^2^) was used to estimate the heterogeneity; this describes the percentage of variation across studies that is due to heterogeneity and not chance [33,34,35].

The Egger’s test was used to assess the publication bias (which occurs when the outcome of a research study influences the publication decision) [33,34,35].

Statistical analyses were performed using StatsDirect software (Version 3, Birkenhead, UK).

## 5. Conclusions

The prevalence of HALNs detected by 2-[^18^F]FDG PET/CT after recent COVID-19 vaccination is significant. Physicians must be aware and recognize the possibility of HALNs related to immune response to vaccine injection in the setting of mass vaccination of the population to avoid patient anxiety and unnecessary further examinations or biopsies. Larger studies are needed to confirm the findings of this systematic review and meta-analysis.

## Figures and Tables

**Figure 1 pharmaceuticals-14-00762-f001:**
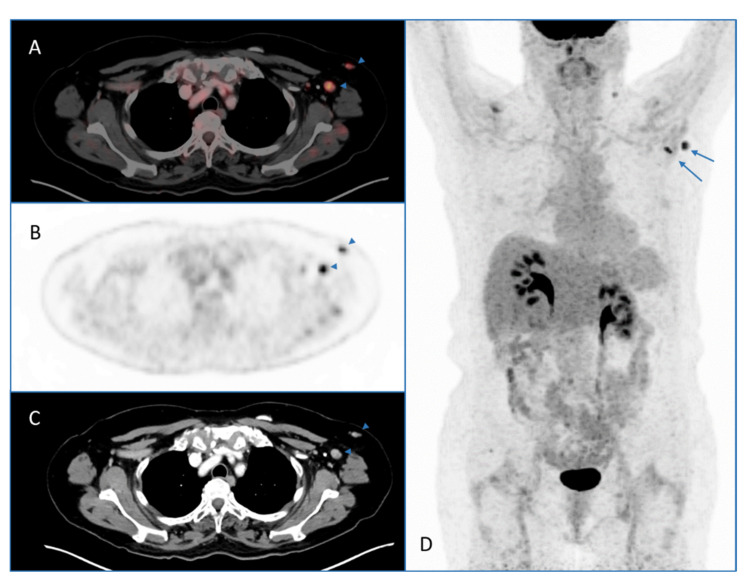
2-[^18^F]FDG PET/CT performed for oncological indication in a patient with previous COVID-19 vaccination (3 weeks before 2-[^18^F]FDG PET/CT). Axial PET/CT (**A**), PET (**B**), and CT (**C**) images and maximum intensity projection (MIP) PET image (**D**) showed hypermetabolic axillary lymph nodes due to increased 2-[^18^F]FDG uptake in the left axillary region (arrows). These findings were judged as reactive lymph nodes after COVID-19 vaccination.

**Figure 2 pharmaceuticals-14-00762-f002:**
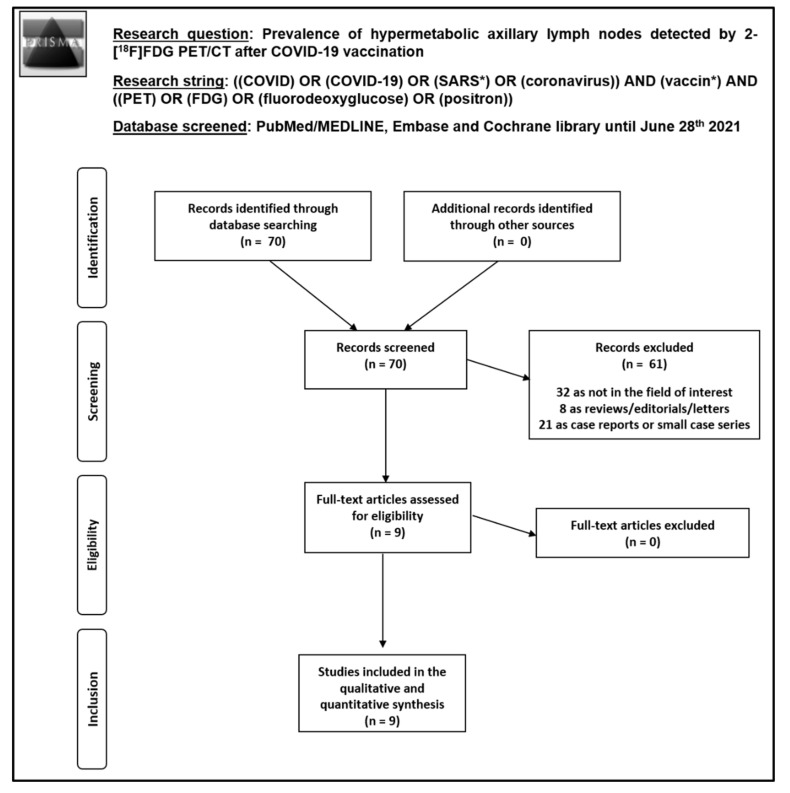
Process of article selection.

**Table 1 pharmaceuticals-14-00762-t001:** Basic study and patient characteristics.

Authors	Country	Study Design	Type of Patients Evaluated	No. of Patients Evaluated with 2-[^18^F]FDG PET/CT	Mean Age	Male %	Type of COVID-19 Vaccine (Manufacturer)	2-[^18^F]FDG PET/CT after First Dose of COVID-19 Vaccine	2-[^18^F]FDG PET/CT after Second Dose of COVID-19 Vaccine	Time between COVID-19 Vaccination and PET/CT Scan (Days)
Adin et al. [10]	USA	R	Patients with previous recent COVID-19 vaccination who underwent 2-[^18^F]FDG PET/CT for oncological or other indications	68	75	47%	mRNA vaccine (Moderna and Pfizer)	41 (60%)	27 (40%)	1–47
Bernstine et al. [11]	Israel	R	Patients with previous recent COVID-19 vaccination who underwent 2-[^18^F]FDG PET/CT for oncological indications	650	69	46%	mRNA vaccine (Pfizer)	394 (61%)	256 (39%)	NR
Cohen et al. [12]	Israel	R	Patients with previous recent COVID-19 vaccination who underwent 2-[^18^F]FDG PET/CT for oncological or other indications	728	69	43%	mRNA vaccine (Pfizer)	346 (48%)	382 (52%)	NR
Cohen et al. [13]	Israel	R	Patients with previous recent COVID-19 vaccination who underwent 2-[^18^F]FDG PET/CT for evaluation of hematological malignancy	137	68.5	55%	mRNA vaccine (Pfizer)	51 (37%)	86 (63%)	5–30
Eifer et al. [14]	Israel	R	Patients with previous recent COVID-19 vaccination who underwent PET/CT with several radiotracers for oncological or other indications	377	67	51%	mRNA vaccine (Pfizer)	301 (80%)	76 (20%)	1–34
Eshet et al. [15]	Israel	R	Patients with previous COVID-19 vaccination who underwent 2-[^18^F]FDG PET/CT for oncological or other indication beyond 6 weeks after vaccination	169	65	51%	mRNA vaccine (Pfizer)	0	169 (100%)	42–71
Schroeder et al. [16]	USA	R	Patients with previous recent COVID-19 vaccination who underwent 2-[^18^F]FDG or radiolabeled choline PET/CT for oncological indications	54	76	64%	mRNA vaccine (Moderna and Pfizer)	NR	NR	1–42
Shin et al. [17]	Korea	R	Healthy subjects with previous recent COVID-19 vaccination who underwent 2-[^18^F]FDG PET/CT for cancer screening	31	45	35%	Adenovirus-vectored vaccine (AstraZeneca)	NR	NR	1–29
Skawran et al. [18]	Switzerland	R	Patients with previous recent COVID-19 vaccination who underwent 2-[^18^F]FDG PET/CT for oncological indications	140	67	72%	mRNA vaccine (Moderna and Pfizer)	48 (34%)	92 (66%)	0–48

Legend: 2-[^18^F]FDG = fluorine-18 fluorodeoxyglucose; mRNA = messenger ribonucleic acid; NR = not reported; PET/CT = positron emission tomography/computed tomography; R = retrospective.

**Table 2 pharmaceuticals-14-00762-t002:** Technical aspects of 2-[^18^F]FDG PET/CT in the included studies.

Authors	Hybrid Imaging Modality and PET/CT Scanner	Fasting before 2-[^18^F]FDG Injection	Mean Injected 2-[^18^F]FDG Activity	Time Interval between 2-[^18^F]FDG Injection and Image Acquisition	Image Analysis
Adin et al. [10]	NR	4–6 h	NR	1 h	Visual
Bernstine et al. [11]	PET/CT (contrast enhanced CT) using GE Discovery 710	NR	185–370 MBq	NR	Visual and semi-quantitative (SUV_max_)
Cohen et al. [12]	PET/CT (contrast enhanced CT) using GE Discovery 690 or GE Discovery MI	NR	3.7 MBq/kg	1 h	Visual and semi-quantitative (SUV_max_)
Cohen et al. [13]	PET/CT (contrast enhanced CT) using GE Discovery 690 or GE Discovery MI	NR	3.7 MBq/kg	1 h	Visual and semi-quantitative (SUV_max_)
Eifer et al. [14]	PET/CT (low dose CT) using Philips Vereos	2–6 h	5.18 MBq/kg	1 h	Visual and semi-quantitative (SUV_max_)
Eshet et al. [15]	PET/CT (low dose CT) using Philips Vereos	NR	NR	NR	Visual and semi-quantitative (SUV_max_)
Schroeder et al. [16]	PET/CT (low dose CT) using GE Discovery 690, GE Discovery 710, GE Discovery MI or Siemens Biograph Vision 600	11 h	437 MBq	1 h	Visual and semi-quantitative (SUV_max_)
Shin et al. [17]	PET/CT (low dose CT) using GE Discovery STE	At least 6 h	5 MBq/kg	1 h	Visual and semi-quantitative (SUV_max_)
Skawran et al. [18]	PET/CT (low dose CT) using GE Discovery MI	At least 4 h	1.5–3.1 MBq/kg	1 h	Visual and semi-quantitative (SUV_max_)

Legend: 2-[^18^F]FDG = fluorine-18 fluorodeoxyglucose; CT = computed tomography; h = hour; MBq = megabecquerel; NR = not reported; PET/CT = positron emission tomography/computed tomography; SUV_max_ = maximal standardized uptake value.

**Table 3 pharmaceuticals-14-00762-t003:** Data on the prevalence of hypermetabolic axillary lymph nodes and deltoid muscle uptake at 2-[^18^F]FDG PET/CT after COVID-19 vaccine.

Authors	All Cases Evaluated with 2-[^18^F]FDG PET/CT after COVID-19 Vaccine			Cases Evaluated with 2-[^18^F]FDG PET/CT after the First Dose of COVID-19 Vaccine		Cases Evaluated with 2-[^18^F]FDG PET/CT after the Second Dose of COVID-19 Vaccine	
	HALNs Present	HALNs Absent	Uptake at Injection Site of COVID-19 Vaccine	HALNs Present	HALNs Absent	HALNs Present	HALNs Absent
Adin et al. [10]	9/68 (13%)	59/68 (87%)	8/68 (12%)	2/41 (5%)	39/41 (95%)	7/27 (26%)	20/27 (74%)
Bernstine et al. [11]	168/650 (26%)	482/650 (74%)	52/168 (31%)	57/394 (14%)	337/394 (86%)	111/256 (43%)	145/256 (57%)
Cohen et al. [12]	332/728 (46%)	396/728 (54%)	99/266 (37%)	126/346 (36%)	220/346 (64%)	206/382 (54%)	176/382 (46%)
Cohen et al. [13]	43/137 (31%)	94/137 (69%)	NA	13/51 (25%)	38/51 (75%)	30/86 (35%)	56/86 (65%)
Eifer et al. [14]	170/377 (45%)	207/377 (55%)	98/377 (26%)	NA	NA	NA	NA
Eshet et al. [15]	49/169 (29%)	120/169 (71%)	NA	NA	NA	49/169 (29%)	120/169 (71%)
Schroeder et al. [16]	4/54 (7%)	50/54 (93%)	8/55 (15%)	NA	NA	NA	NA
Shin et al. [17]	28/31 (90%)	3/31 (10%)	22/30 (73%)	NA	NA	NA	NA
Skawran et al. [18]	75/140 (54%)	65/140 (46%)	NA	27/48 (56%)	21/48 (44%)	48/92 (52%)	44/92 (48%)
**Pooled values (95%CI)**	37% (27–47%)	63% (53–73%)	30% (20–41%)	26% (13–42%)	74% (58–87%)	41% (32–50%)	59% (50–68%)
**Heterogeneity (I^2^)**	High (95%)	High (95%)	High (90%)	High (95%)	High (95%)	High (87%)	High (87%)
**Egger’s test (publication bias)**	*p* = 0.8 (absent)	*p* = 0.8 (absent)	*p* = 0.6 (absent)	*p* = 0.5 (absent)	*p* = 0.5 (absent)	*p* = 0.3 (absent)	*p* = 0.3 (absent)

Legend: 2-[^18^F]FDG = fluorine-18 fluorodeoxyglucose; 95%CI = 95% confidence interval; HALNs = hypermetabolic axillary lymph nodes; NA = not available or calculable; PET/CT = positron emission tomography/computed tomography.

**Table 4 pharmaceuticals-14-00762-t004:** Data on the prevalence of hypermetabolic lymph nodes at 2-[^18^F]FDG PET/CT after injection of other vaccines beyond COVID-19.

Authors and Year	Target of Vaccination	Time Interval from Vaccine Injection to 2-[^18^F]FDG PET/CT Scan (Days)	Cases Evaluated with 2-[^18^F]FDG PET/CT after Vaccination		
			HyperMetabolic LN Present	HyperMetabolic LN Present	Uptake at Injection Site of Vaccine
Burger et al. 2011 [22]	Influenza	1–30	17/58 (29%)	41/58 (71%)	17/58 (29%)
Coates et al. 2017 [23]	Papillomavirus	8–37	15/15 (100%)	0/15 (0%)	NA
Iyenga et al. 2003 [24]	Influenza	3–5	7/8 (87%)	1/8 (13%)	NA
Nakata et al. 2021 [25]	Anti-cancer	1–1159	NA	NA	33/37 (89%)
Panagiotidis et al. 2010 [26]	Influenza	2–18	10/10 (100%)	0/10 (0%)	NA
Shirone et al. 2012 [27]	Influenza	<7 or ≥7	4/83 (5%)	79/83 (95%)	NA
Thomassen et al. 2011 [28]	Influenza	1–330	NA	NA	NA
Win et al. 2021 [29]	Several types of viruses	1–10	38/53 (72%)	15/53 (28%)	NA

Legend: 2-[^18^F]FDG = fluorine-18 fluorodeoxyglucose; 95%CI = 95% confidence interval; LN = lymph nodes; NA = not available or calculable; PET/CT = positron emission tomography/computed tomography.

## Data Availability

Data sharing not applicable.

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
