# Peer review of "Prevalence and Significance of Hypermetabolic Lymph Nodes Detected by 2-[18F]FDG PET/CT after COVID-19 Vaccination: A Systematic Review and a Meta-Analysis"

_pharmaceuticals, 2021, doi:10.3390/ph14080762_

Round 1

Reviewer 1 Report

This is a detailed and quite interesting meta-analysis on FDG PET/CT findings after covid vaccine.

The manuscript is well concepted, properly structured and clearly written

Main limitations are those already expressed by the authors, in particular the absence of histologic proof, anyway as per meta-analysis nature, the aim of this article is to describe specific evidences in literature.

I do not have major findings.

Only minor comments:

  • In my opinion it woul be more interesting and useful for clinicians to deepen radiologic findings in oncologic patients that are those more at risk of PET-findings misinterpretation
  • Some more details on morphologic co-registration CT scan findings should be provided
  • references to existing works on morphologic evaluation of post-vaccine lymphoadenopathies (e.g. ultrasound, CT scan) should be included

Author Response

Thank you for your comments on our manuscript.

We did not focus our systematic review and meta-analysis on CT findings; the reason is that HALNs are frequently normal in size at CT scan. However, we would like to underline that it is important that radiologists consider recent COVID-19 vaccination in the differential diagnosis of unilateral axillary lymphadenopathy and are aware of typical appearances of this findings across all imaging methods.

We have added this statement and related references in the discussion of the revised manuscript.

Reviewer 2 Report

This is an interesting systematic review of the use of PET-CT for the detection and differential diagnosis of HALNs. The authors have conducted a thorough research in the most important biomedical databases. The methods are well described and the analysis has been conducted lege artis. The results are also well described and important, since this vaccine side effect is alarming especially in cancer patients. Minor editing is needed for grammatical errors and typos. 

Author Response

Thank you for your comments on our article.

We performed the minor revision requested.